# Ensemble Three-Dimensional Habitat Modeling of Indian Ocean Immature Albacore Tuna (*Thunnus alalunga*) Using Remote Sensing Data

Sandipan Mondal [1], Yi-Chen Wang [1,2], Ming-An Lee [3,4,*], Jinn-Shing Weng [5] and Biraj Kanti Mondal [6]

1 Department of Environmental Biology Fisheries Science, National Taiwan Ocean University, No. 2, Beining Rd., Zhongzheng Dist., Keelung City 20224, Taiwan
2 Center of Excellence for Ocean Engineering, National Taiwan Ocean University, Keelung City 20224, Taiwan
3 Doctoral Degree Program in Ocean Resource and Environmental Changes, National Taiwan Ocean University, No. 2, Beining Rd., Zhongzheng Dist., Keelung City 20224, Taiwan
4 Center of Excellence for Oceans, National Taiwan Ocean University, No. 2, Beining Rd., Zhongzheng Dist., Keelung City 20224, Taiwan
5 Coastal and Offshore Resources Research Center of Fisheries, Research Institute, Kaoshiung City 80672, Taiwan
6 Department of Geography, School of Sciences, Netaji Subhas Open University, Kolkata 700064, West Bengal, India
* Correspondence: malee@mail.ntou.edu.tw; Tel.: +886-2-24622192

**Abstract:** This study evaluated the vertical distribution of immature albacore tuna (*Thunnus alalunga*) in the Indian Ocean as a function of various environmental parameters. Albacore tuna fishing data were gathered from the logbooks of large-sized Taiwanese longline vessels. Fishery and environmental data for the period from 1998 to 2016 were collected. In addition to the surface variable, the most influential vertical temperature, dissolved oxygen (OXY), chlorophyll, and salinity layers were found at various depths (i.e., 5, 26, and 53 m for SST; 200, 244, and 147 m for OXY; 508, 628, and 411 for SSCI; and 411, 508, and 773 m for SSS) among 20 vertical layers based on Akaike criterion information value of generalized linear model. Relative to the 20 vertical layers base models, these layers had the lowest Akaike information criteria. For the correlation between the standardized and predicted catch per unit effort (CPUE), the correlation values for the generalized linear model (GLM), generalized additive model (GAM), boosted regression tree (BRT), and random forest (RF) model were 0.798, 0.832, 0.841, and 0.856, respectively. The GAM-, BRT-, and RF-derived full models were selected, whereas the GLM-derived full model was excluded because its correlation value was the lowest among the four models. From March to September, a higher immature albacore standardized CPUE was mainly observed from 30°S to 40°S. A northward shift was observed after September, and the standardized CPUE was mainly concentrated at the south coast of Madagascar from November to January.

**Keywords:** albacore; ensemble species modelling; salinity; temperature; vertical distribution

## 1. Introduction

Albacore tuna is a highly migratory and carnivorous species belonging to the Scombridae family [1]. This species is found in the temperate waters of all the three major oceans (Indian, Pacific, and Atlantic) and is cosmopolitan in nature [2–4]. Albacore tuna is a key commercial species, accounting for up to 6% of total global tuna catches by weight [5]. Taiwanese longline vessels have been fishing for Indian Ocean albacores since 1950. Taiwanese drifting longline vessels account for the majority of albacore catches in the Indian Ocean (90% of total catches). The output of albacore tuna has increased in response to its growing demand. The albacore harvest achieved by Taiwanese longline vessels increased from 10,000 tons in 1950 to 102,594 tons in 2018 [6,7]. However, high exploitation levels ca

lead to overfishing, which can cause the collapse of albacore stocks (recovery may require several decades) and threaten livelihoods, communities, and food security. Therefore, the spatial distribution of this species in the Indian Ocean must be comprehensively evaluated to identify potential higher or lower habitat areas on the basis of marine habitat data. Sustainable fishing- and ecosystem-based fishery management can be implemented by regulating fishing efforts in higher or lower habitat regions.

Species distribution modeling (SDM; also known as habitat modeling, ecological niche modeling, bioclimatic envelope modeling, or resource selection function modeling [8–10]) is the most frequently used technique for evaluating the habitat pattern of a species. The basic premise of habitat modeling is to estimate species distribution by using computer algorithms based on mathematical representations of the known distribution of a species in environmental regions [11]. Reality, generality, and precision are the three main objectives of habitat modeling [12]. Numerous statistical relationships between existing species distributions and environmental variables are used in the prediction analysis.

To examine the potential effect of environmental variability on the distribution of a species, high-resolution spatiotemporal oceanographic data are required. This type of information can be found in data assimilation model products provided by several satellites, including the Moderate Resolution Imaging Spectroradiometer (MODIS), Copernicus (COP), and the Advanced Very High-Resolution Radiometer (AVHRR) etc., which provide information on vertical oceanographic structures and how they affect species distribution. The basic difference between different satellites is their spatial resolution. Multisatellite detection has been performed to collect data on various oceanographic parameters since 1978. These data are particularly beneficial in the fields of oceanography and fishery management [13–15]. Because of the availability of large-scale data, a further analysis can enable obtaining useful information on fisheries misuse and management [16–19]. Multisatellite detection has increased our understanding of factors that affect the living habitats of fish species and other related species [20–24]. In addition, data obtained through precise detection can help academicians to develop cost-effective fishery management models, modelers to generate statistics, and fishermen to become more fuel-efficient when attempting to locate fishing sites [25].

Model comparison studies have revealed that various modeling methods differ in terms of their statistical and predictive performance [26–29]. This might be due to basic differences in model complexity. Researchers have examined multimodel frameworks for obtaining robust forecasts by combining the capabilities of multiple model algorithms that apply an ensemble model forecast technique [30,31]. To produce forecasts ensembles, several simulations spanning multiple sets of initial conditions, model classes, parameters, and boundary conditions are used [32]. A combination of forecasts produces a lower mean error than any of its constituent single projections, when thees projection contains information on single parameters [33]. Several recent studies have applied the ensemble model technique for maritime habitat mapping and reported that this technique allows the merging of habitat projections from multiple model algorithms with decreased bias and high predictive accuracy [30,31,34].

By using the ensemble model technique and three-dimensional oceanographic data, we evaluated the probable habitats of immature albacore tuna in the Indian Ocean. In our previous study, only a single-algorithm model and surface oceanographic factors were used to analyze immature albacore (average weight of <14 kg) habitat in the study region [35]. Although other habitat model studies have produced limited immature albacore habitat predictions, we could not evaluate preferable vertical immature albacore habitat characteristics because of the availability of numerous modeling platforms. Thus, the present study examined the geographical and temporal patterns of immature albacore habitat by using weighted mean ensemble projections (obtained from species distribution models) and three-dimensional ocean data to fill potential information gaps. Thus, the present study explored robust three-dimensional immature albacore habitat characteristics derived from an ensemble of model forecasts and investigated the mechanistic linkage between the

spatial and temporal variability of potential immature albacore habitats under changing environmental conditions. Figure 1 illustrates the experimental flow of the present study. For all the acronyms, refer to Abbreviations Section.

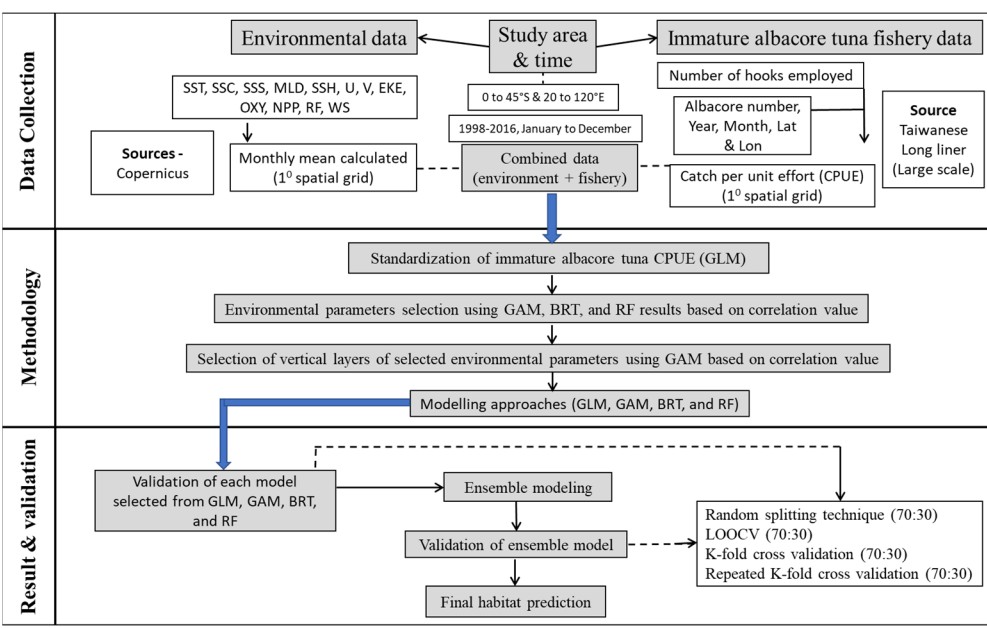

**Figure 1.** Experimental flowchart.

## 2. Materials and Methods

### 2.1. Data Collection

#### 2.1.1. Albacore Tuna Fishery Data

Weight at first maturity of Indian Ocean albacore is 14 kg [5]. Following this, an average weight of 14 kg was taken as the threshold to separate mature and immature albacore in the present study. The present study divided CPUE data into two categories, which represent different life history stages of albacore tuna: immature (average weight <14 kg), mature (average weight >14 kg). Only immature albacore data were used in the present study (as present study is the continuation of Mondal et al., 2021).

Immature albacore tuna fishery data for the period from 1998 to 2016 were collected from the fishing logbooks of large-sized longline vessels (deep-water fishing vessels with a registered tonnage of >100 tons and a length of >24 m) of the Overseas Fisheries Development Council of Taiwan. Data from small-sized fishing vessels (mostly coastal water fishing vessels with a gross registered tonnage of <100 tons and a length of <24 m) were not used in the present study because of the lack of data pertaining to the study period. The data have a spatial coverage from 0°S to 45°S and from 20°E to 120°E with a spatial resolution of 1° × 1°. The logbooks recorded the year, month, latitude, longitude, number of catches, number of hooks used, hooks per basket (data were not available for specific years), and weight (whether the weight was dry or wet weight was not specified). Data related to soaking time, hook depth, and operation time were not found in the data set.

#### 2.1.2. Oceanographic Data

For our analysis, data on 10 oceanographic parameters were acquired from Copernicus (The GLORYS12V1 product is the CMEMS global ocean eddy-resolving with 1/12° horizontal resolution from ERA5 reanalysis covering the altimetry (Table 1). The model component is the NEMO. Along with track altimeter data, satellite data of vertical profiles of different environmental parameters are jointly assimilated with a processing level of L4. The data covered a range of spatial resolutions from 0°S to 45°S (albacore fishing is only conducted in this region during specific months) and from 20°E to 120°E. To match

the large-scale albacore tuna fisheries statistics from 1998 to 2016, data from all months (i.e., January to December) for the period from 1998 to 2016 were collected. Environmental data did not have a spatial coverage of $1° × 1°$; these data were interpolated to a $1° × 1°$ spatial grid (also the daily data were converted to monthly data) using MATLAB version 2019a (Kriging method—Kriging is a geostatistics method that predicts the value in a geographic area given a set of measurements) because the spatial resolution of the fisheries data was $1° × 1°$. As of February 2, 2022, all data were accessible. Data with daily temporal resolution were converted into a monthly base using MATLAB version 2019a because the temporal resolution of fisheries data was monthly. An earlier study [4] mentioned that a phytoplankton patch must mature into a foraging ground after a minimum of 5–7 days, and it was suggested that 1-month lag chlorophyll data be used as a potential predictor. Additionally, it could take some time to find the predating zone by searching, therefore a larger concentration of SSC might not always indicate a larger albacore biomass at a given moment. In order to determine whether there was any meaningful justification for employing SSC lag data, the authors attempted to use the lag data of SSC as well.

**Table 1.** Sources of various oceanographic variables derived from satellite source.

| Environmental Data | Abb. | Unit | Source | Time Period | Spatial Resolution | Temporal Resolution |
|---|---|---|---|---|---|---|
| Temperature | SST | °C | COP | 1998–2016 | $0.08° × 0.08°$ | Monthly |
| Dissolved oxygen | OXY | mmol/L | COP | 1998–2016 | $0.08° × 0.08°$ | Monthly |
| Chlorophyll (0–2 months lag) | SSC (0–2) | $mgm^{-3}$ | COP | 1998–2016 | $0.25° × 0.25°$ | Monthly |
| Salinity | SSS | psu | COP | 1998–2016 | $0.08° × 0.08°$ | Monthly |
| U-velocity | U | $ms^{-1}$ | COP | 1998–2016 | $0.08° × 0.08°$ | Monthly |
| V-velocity | V | $ms^{-1}$ | COP | 1998–2016 | $0.08° × 0.08°$ | Monthly |
| Eddie kinetic energy | EKE | $m^2s^{-2}$ | COP | 1998–2016 | $0.08° × 0.08°$ | Monthly |
| Net primary productivity | NPP | $mgm^{-3}day^{-1}$ | COP | 1998–2016 | $0.25° × 0.25°$ | Monthly |
| Mixed layer depth | MLD | meter | COP | 1998–2016 | $0.08° × 0.08°$ | Monthly |
| Sea surface height above geoid | SSH | meter | COP | 1998–2016 | $0.08° × 0.08°$ | Daily |

COP—Copernicus. (https://resources.marine.copernicus.eu/products, accessed on 14 August 2022) EKE = 0.5 $(U^2 + V^2)$. Unit for EKE is $m^2s^{-2}$.

### 2.2. Standardization of Nominal Catch Per Unit Effort

The relative abundance of immature albacore was indexed as catch per unit effort (CPUE, nominal). Nominal CPUE (N.CPUE; per 1000 hooks) was calculated using the following formula:

$$\text{N.CPUE} = (\text{No. of albacore catch})/(\text{No. of hooks used}) \tag{1}$$

In fisheries, CPUE is used as a reliable proxy for relative abundance. However, using raw CPUE as an index of abundance can be problematic sometime. This is because CPUE can be "hyper stable" or less sensitive to the rapid changes of abundance. Changes in the fishing location, strategy, season, and fishing pattern can cause changes to CPUE that are independent of relative abundance. To reduce the dominance of several spatial (latitude and longitude) and temporal (year and month) factors, N.CPUE was standardized using the common method of generalized linear modeling (GLM) to obtain a bias-filtered data set (standardized CPUE, S.CPUE). The mgcv package [11,19,22] was used to build a stepwise GLM model (Gaussian distribution) with five factors (Year, month, latitude, longitude, and interactions) in R-studio version 3.6.0. Three interactions i.e., Year*Lat, Year*Lon, and Lat*Lon were all included together. A total of five models were examined, and the optimal model for standardization was selected on the basis of the lowest Akaike information criterion (AIC) [4], the most deviance explained (percentage), and highest correlation ($R^2$) values. The Akaike information criterion (AIC) is an estimator of prediction error, and thereby relative quality of statistical models, for a given set of data. Given a collection of

models for the data, AIC estimates the quality of each model, relative to each of the other models. The GLM models were built using the following formula:

$$\text{GLM}_n: \text{Log (N.CPUE} + c) \sim a_1 + a_2 + a_3 + \ldots + a_n + \mu + \text{€} \qquad (2)$$

where c is the constant value of 0.1, *n* is the number of variables, $\text{GLM}_n$ is the model with *n* factors, μ is the interaction (Year*Lat, Year*Lon, and Lat*Lon), and € is a variable with normal distribution and a mean value of zero.

### 2.3. Selection of Oceanographic Parameters and Vertical LAYER

The generalized additive model (GAM) is often used to select suitable environmental parameters prior to the development of a habitat model [35,36]. Because of complex relationships between an angle community and the environment, determining whether the relationship is linear or nonlinear is a difficult task. Therefore, models such as GAM (Gaussian distribution with smoothing spline regression), which allow for nonlinear reactions, are more suitable than other models for examining relationships between angle communities and conditions [37–41]). The use of boosted regression trees (BRTs) (no. of regression trees used was 531 with interaction depth of 7 and bag fraction of 0.6) and random forests (RFs) (no. of regression trees used was 325 with interaction depth of 4 and bag fraction of 0.8) can aid parameter selection. Nonparametric random forest models can model highly nonlinear relationships, resulting in improved classification performance. A BRT is a type of a nonlinear model that divides data into regions on the basis of if–then questions. The three aforementioned methods were employed in the present study to select appropriate parameters, and each parameter was rated using each method on the basis of their correlation value. Only parameters with a Pearson correlation value of >0.3 for at least two of the three methods (i.e., GAM, BRT, and RF) were selected for our subsequent analysis. Under the shaky linear rule, a correlation value of <0.3 indicates a weak positive (negative) linear relationship [42]. R studio version 3.6.0 was used to test selected parameters for collinearity. Collinearity between parameter pairs was indicated by a Pearson correlation value of >0.7 [23] and variance inflation factor value of >5.

The most influential oceanographic layers of the selected parameters for immature albacore S.CPUE were identified using a GAM and the mgcv package version 1.8–2 [43]; this method is widely used in the literature [31]. Twenty vertical layers (5, 26, 53, 77, 97, 147, 200, 244, 300, 411, 508, 628, 773, 856, 947, and 1045 m) were initially used, and the vertical profiles for the selected oceanographic variables were subsequently selected on the basis of the three most influential vertical layers [31] that had the lowest AICs [44].

### 2.4. Construction and Evaluation of the Single-Algorithm Habitat Model

Following the finalization of the oceanographic variables, four single-algorithm models were built (full models that had all selected parameters and layers and were validated using BIOMOD2) [45,46]. The four models were the GLM (Gaussian distribution), GAM (Gaussian distribution with smoothing spline regression), BRT (no. of regression trees used were 531 with interaction depth of 7 and bag fraction of 0.6), and RF (no. of regression trees used were 325 with interaction depth of 4 and bag fraction of 0.8). The package's species distribution models also use a set of statistical techniques, which include conventional regression-based models and current machine-learning platforms. Thus, the proposed ranges of the single-algorithm models were all used to investigate and exploit their specific prediction capabilities.

All single-algorithm models were initially assessed using four validation techniques (i.e., random splitting [RS], leave-one-out cross-validation [LOOCV], 10-fold cross-validation [10-fold], and repeated (3) 10-fold cross-validation [repeated 10-fold]. During the validation process, the fishing data set with sample no. of 67,548 was divided into two random parts at a 70:30 (training:testing) ratio. Three coefficients (i.e., Pearson correlation coefficient [R], root-mean-square error [RMSE], and mean absolute error [MAE]) were calculated for both the training and testing data by applying all the validation techniques in each

single-algorithm model. We used the training data set to train the model, while the testing data were used to compare the model performances. 70% of the data were used to train a model. This same model was used to test the rest of the 30% data in order to check the selected model's biasedness for prediction. This biasedness was checked using three coefficients (R, RMSE and MAE). Lesser difference between the coefficient values of training and testing data sets indicated least biasedness. Thus, we applied these methodologies in the present study. Validations were performed using the tidyverse and caret R-packages (version 3.6.0). Smaller differences in R, RMSE, and MAE values between the two data sets (70:30) indicate a more favorable model performance with less bias. After the successful validation, Akaike information criterion [AIC], root-mean-square error [RMSE], and mean absolute error [MAE] were calculated for all single-algorithm final models using 100% data to evaluate the predictive performances of each single-algorithm model. Least AIC, RMSE, and MAE value indicate the best performing model. The single-algorithm model with higher AIC, RMSE, and MAE values was eliminated from the further analysis.

*2.5. Ensemble Model Development, Evaluation, and Prediction*

Following the performance evaluation of the single-algorithm models, we used R, RMSE, and MAE to establish a weighted mean ensemble model for immature albacore habitats by applying the full selected single-algorithm models. The ensemble model was validated using four validation techniques (i.e., RS, LOOCV, 10-fold, and repeated (3) 10-fold). For the validation process, the fishing data set was randomly divided into 2 parts at a 70:30 (training: testing) ratio. Three coefficients (i.e., Pearson correlation coefficient [R], root-mean-square error [RMSE], and mean absolute error [MAE]) were calculated for both the training and testing data by applying all the validation techniques in the ensemble model. Validation was performed using the tidyverse and caret R-packages (version 3.6.0). Lesser differences in R, RMSE, and MAE values between the two data sets (70:30) indicate a more favorable model performance with less bias.

The ensemble model was used to perform the final prediction if the validation method indicated a more favorable model performance with no significant biasedness. The predicted values for each point of the study area from the final model were mapped to a $1° \times 1°$ spatial grid by using the ArcGIS software (version 10.2). The ensemble model was used to predict CPUE (P.CPUE), which was then used as a proxy for relative abundance.

**3. Results**

*3.1. Standardization of Nominal CPUE Data*

The full GLM model (with all the factors) had the lowest AIC value (142,977) and the highest $R^2$ value (0.78) after stepwise GLM model generation was performed (Table 2). The standardization model's quantile–quantile plot and histogram (Figure 2) indicate an almost normal distribution. Therefore, a selected model was employed to standardize the N.CPUE of immature albacore. N.CPUE (monthly summed CPUE) ranged from 0.1 to 4000 (Figure 3). The monthly total CPUE decreased to a range of 0.1 to 2200 after standardization. For the subsequent study of juvenile albacore tuna, S.CPUE was applied.

**Table 2.** Performance of various combinations of generalized linear models for the standardization of immature albacore nominal CPUE.

| No. | Models | AIC | Null | Residual | $R^2$ | P(f) |
|---|---|---|---|---|---|---|
| 1 | Year | 223,937 | 135,722 | 133,006 | 0.002 | <0.001 |
| 2 | Year + Month | 216,436 | 135,722 | 117,864 | 0.131 | <0.001 |
| 3 | Year + Month + Lat | 163,405 | 135,722 | 50,203 | 0.63 | <0.001 |
| 4 | Year + Month + Lat + Lon | 147,018 | 135,722 | 38,471 | 0.716 | <0.001 |
| **5** | **Year + Month + Lat + Lon + Interactions** | **142,977** | **135,722** | **35,249** | **0.781** | **<0.001** |

Lat = Latitude; Lon = Longitude; Interactions = Year*Lat, Year*Lon, Lat*Lon

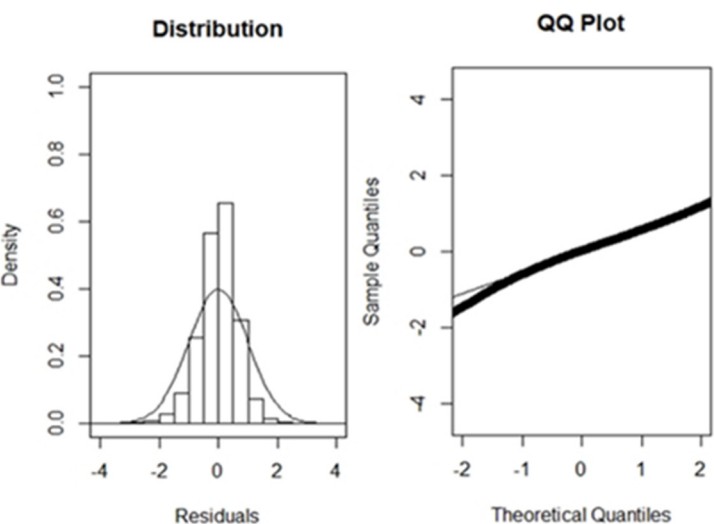

**Figure 2.** Residual distribution and quantile–quantile plot for selected generalized linear models for immature albacore tuna.

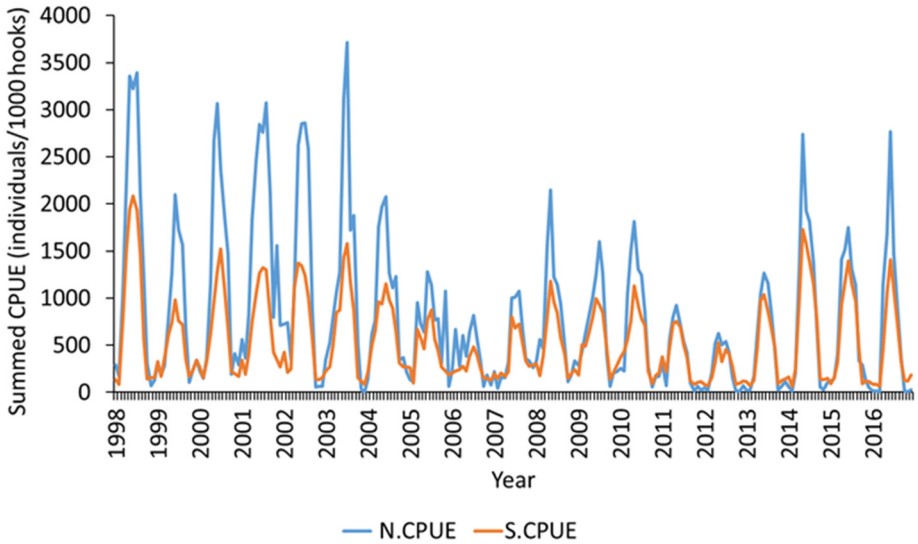

**Figure 3.** Comparison of N.CPUE and S.CPUE using selected generalized linear model for immature albacore tuna.

### 3.2. Oceanographic Parameters and Vertical Layer Selection

Table 3 presents the significance of various environmental characteristics when several selection procedures were applied. The environmental characteristics of dissolved oxygen (OXY), temperature (SST), salinity (SSS), mixed layer depth (MLD), and chlorophyll level with a 1-month lag (SSC1) all had correlation values of >0.3 with S.CPUE when the GAM technique was used. OXY (0.855) had the highest correlation with S.CPUE. The generalized cross-validation (GCV) index for OXY was the lowest among the metrics. For correlation and GCV index values, SSC1 outperformed the chlorophyll level with no lag (SSC0) and chlorophyll level with a 2-month lag (SSC2). OXY, SST, SSS, MLD, SSC1, and eddy kinetic energy (EKE) all had correlation values of >0.3 with S.CPUE when the BRT method was used. OXY (0.857) had the highest correlation with S.CPUE. OXY had the lowest RMSE and MAE index values among all the parameters. In the correlation of RMSE and MAE index values, SSC1 outperformed SSC0 and SSC2. OXY, SST, SSS, MLD, SSC0, SSC1, and SSC2 had correlation values of >0.3 with S.CPUE when the RF technique was used. OXY (0.836) had the highest correlation with S.CPUE. OXY had the lowest RMSE and MAE index values among all the parameters. Although all SSC delays had correlation values of

>0.3, SSC1 outperformed SSC0 and SSC2 in terms of correlation, RMSE, and MAE index values. Consequently, only SSC1 was selected. The aforementioned data indicate that for all the selection methods, OXY, SST, SSS, MLD, and SSC1 had correlation values of >0.3. EKE only had a correlation value of >0.3 when the BRT method was used. Therefore, the final model was constructed using only OXY, SST, SSS, MLD, and SSC1. A multicollinearity test revealed that the selected parameters did not exhibit any collinearity (Table 4).

**Table 3.** Environment parameters selected using GAM, BRT, and RF for immature albacore tuna model construction.

|  | Parameter | AIC | DEV EXP | Adj. R squ. | GCV |
|---|---|---|---|---|---|
| **GAM** | OXY | 86,498.17 | 85.5 | 0.855 | 0.235 |
|  | SST | 86,563.96 | 85.4 | 0.854 | 0.232 |
|  | SSS | 150,073 | 59.6 | 0.596 | 0.652 |
|  | MLD | 174,173.6 | 40.5 | 0.405 | 0.961 |
|  | SSC1 | 179,620.9 | 36.1 | 0.361 | 1.049 |
|  | SSC0 | 185,294.5 | 28.9 | 0.289 | 1.149 |
|  | SSC2 | 195,414.9 | 21.3 | 0.213 | 1.912 |
|  | EKE | 186,516.8 | 27.5 | 0.274 | 1.721 |
|  | U | 187,736.4 | 26 | 0.26 | 1.195 |
|  | V | 199,962.5 | 9.96 | 0.099 | 1.454 |
|  | SSH | 200,105.2 | 9.75 | 0.097 | 1.458 |
|  | NPP | 205,861.6 | 1.01 | 0.009 | 1.599 |

|  | Parameter | RMSE | DEV EXP | Adj. R squ. | MAE |
|---|---|---|---|---|---|
| **BRT** | OXY | 0.48 | 85.7 | 0.857 | 0.255 |
|  | SST | 0.483 | 85.5 | 0.855 | 0.264 |
|  | SSS | 0.797 | 60.5 | 0.605 | 0.462 |
|  | MLD | 0.979 | 40.5 | 0.405 | 0.662 |
|  | SSC1 | 0.992 | 39.4 | 0.394 | 0.678 |
|  | SSC0 | 1.123 | 27.9 | 0.279 | 0.84 |
|  | SSC2 | 1.145 | 24.5 | 0.245 | 0.856 |
|  | EKE | 1.002 | 32.8 | 0.328 | 0.691 |
|  | U | 1.077 | 28.4 | 0.284 | 0.789 |
|  | V | 1.199 | 10.9 | 0.109 | 0.942 |
|  | SSH | 1.206 | 9.8 | 0.098 | 0.904 |
|  | NPP | 1.242 | 4.7 | 0.047 | 0.991 |

|  | Parameter | RMSE | DEV EXP | Adj. R squ. | MAE |
|---|---|---|---|---|---|
| **RF** | OXY | 0.445 | 83.6 | 0.836 | 0.251 |
|  | SST | 0.478 | 83.1 | 0.831 | 0.259 |
|  | SSS | 0.717 | 62.1 | 0.621 | 0.465 |
|  | MLD | 0.953 | 42.4 | 0.424 | 0.657 |
|  | SSC1 | 0.987 | 37.1 | 0.371 | 0.673 |
|  | SSC0 | 1.005 | 32.7 | 0.327 | 0.684 |
|  | SSC2 | 1.021 | 30.9 | 0.309 | 0.721 |
|  | EKE | 1.138 | 25.1 | 0.251 | 0.87 |
|  | U | 1.179 | 23.3 | 0.233 | 0.859 |
|  | V | 1.087 | 21.2 | 0.212 | 0.752 |
|  | SSH | 1.234 | 12.3 | 0.123 | 0.935 |
|  | NPP | 1.241 | 9.7 | 0.097 | 0.898 |

A GAM analysis of the effects of vertical SST, OXY, SSC1, and SSS on immature albacore S.CPUE was performed to identify the subsurface habitat characteristics of immature albacore. In addition to the surface variable (MLD), the most influential vertical SST, OXY, SSC1 and SSS layers were found at various depths (i.e., 5, 26, and 53 m for SST; 200, 244, and 147 m for OXY; 508, 628, and 411 for SSC1; and 411, 508, and 773 m for SSS). Relative to the upper-surface base models, these layers had lower AICs. Table 5 evaluated the different selected environmental layers, with respect to S.CPUE, based on Pearson

correlation analysis and VIP score ranking. Results showed that temperature at 5 m depth was more important than other selected layers, followed by temperature at 26 m depth. Temperature at 53 m depth did not show any significant importance, with a VIP rank of 7. The third most important was oxygen at 147 m depth, followed by 200 m and 244 m. For both the SSS and SSC1, 508 m depth showed more importance than 411 m and 628 m depth. MLD showed the lowest correlation with S.CPUE, with a VIP ranking of 13.

**Table 4.** (**a**) Multicollinearity test results for parameters selected for the immature albacore tuna model based on Pearson correlation analysis value. (**b**) Multicollinearity test results for parameters selected for the immature albacore tuna model based on VIF value.

| | OXY | SST | SSS | MLD | SSC1 |
|---|---|---|---|---|---|
| | | | **(a)** | | |
| **OXY** | 1 | | | | |
| **SST** | −0.58 | 1 | | | |
| **SSS** | 0.48 | −0.53 | 1 | | |
| **MLD** | 0.55 | −0.56 | 0.4 | 1 | |
| **SSC1** | 0.3 | −0.42 | 0.16 | 0.3 | 1 |

| | OXY | SST | SSS | MLD |
|---|---|---|---|---|
| | | | **(b)** | |
| **OXY** | | | | |
| **SST** | 4.3 | | | |
| **SSS** | 3.8 | 4.0 | | |
| **MLD** | 4.1 | 4.3 | 3.7 | |
| **SSC1** | 2.7 | −3.5 | 1.1 | 2.9 |

**Table 5.** Evaluation of different selected environmental layers with respect to S.CPUE.

| Parameters | R-squ. | VIP |
|---|---|---|
| OXY_200 | 0.71 | 4 |
| OXY_244 | 0.70 | 5 |
| OXY_147 | 0.73 | 3 |
| TEM_5 | 0.76 | 1 |
| TEM_26 | 0.75 | 2 |
| TEM_53 | 0.69 | 7 |
| SAL_508 | 0.71 | 6 |
| SAL_628 | 0.68 | 8 |
| SAL_411 | 0.62 | 11 |
| SSC1_508 | 0.66 | 9 |
| SSC1_628 | 0.67 | 10 |
| SSC1_411 | 0.60 | 12 |
| MLD | 0.49 | 13 |

*3.3. Relation between Selected Environmental Layers & S.CPUE*

Table 6 showed the optimal ranges of different environmental parameters in different vertical layers with respect to the highest S.CPUE. The optimal range of OXY for both_ 200, and 147 m depth was 240–260 mmol/L with respect to the highest S.CPUE, whereas for the 244 m depth this range was a little lesser (235–255). The optimal range of TEM at 5, 26, and 53 m depth was 13–15 °C, 12–14 °C, and 14–16 °C, respectively, for the highest S.CPUE. The optimal range of SSS for both_508, and 628 m depth was 34.3–34.4 psu, whereas this range was a little higher at 411 m depth (34.4–34.5 psu). For MLD, the optimal range was between 240–260 m for the highest S.CPUE.

**Table 6.** Optimal ranges of different environmental parameters in different vertical layers with respect to the highest S.CPUE.

| Parameters | Depth | Optimal Range | Units | S.CPUE |
|---|---|---|---|---|
| OXY | 200 | 240–260 | mmol/L | 12.17 |
| | 244 | 235–255 | | 11.38 |
| | 147 | 240–260 | | 11.99 |
| TEM | 5 | 13–15 | °C | 12.24 |
| | 26 | 12–14 | | 11.91 |
| | 53 | 14–16 | | 10.63 |
| SSS | 508 | 34.3–34.4 | psu | 10.52 |
| | 628 | 34.3–34.4 | | 10.14 |
| | 411 | 34.4–34.5 | | 8.33 |
| SSC1 | 508 | 0.012–0.013 | $Mgm^{-3}$ | 15.34 |
| | 628 | 0.005–0.006 | | 7.86 |
| | 411 | 0.02–0.021 | | 19.466 |
| MLD | | 250–260 | meter | 19.14 |

*3.4. Predictive Performance of Single-Algorithm Habitat Models*

All the single-algorithm full models exhibited favorable predictive performance. The S.CPUE–P.CPUE Pearson correlation values obtained using the GLM, GAM, BRT, and RF methods were 0.798, 0.832, 0.841, and 0.856, respectively, from Pearson correlation analysis. Table 7 presents the prediction performance (as measured by RMSE, MAE, and $R^2$) achieved using the full single-algorithm models for the selected parameters when each of the three validation techniques were used. The RMSE, MAE, and $R^2$ values obtained from the randomly split data sets exhibited no significant bias in predictive performance. All the single-algorithm models were revealed to be suitable for making predictions.

**Table 7.** Prediction performance of single-algorithm models when three validation techniques were applied.

| Validation Techniques | Methods | RMSE | | $R^2$ | | MAE | |
|---|---|---|---|---|---|---|---|
| | | 70 | 30 | 70 | 30 | 70 | 30 |
| 10 fold | GLM | 0.516 | 0.507 | 0.819 | 0.809 | 0.388 | 0.381 |
| | GAM | 0.521 | 0.514 | 0.817 | 0.807 | 0.384 | 0.378 |
| | BRT | 0.515 | 0.501 | 0.818 | 0.811 | 0.385 | 0.382 |
| | RF | 0.514 | 0.502 | 0.818 | 0.813 | 0.387 | 0.383 |
| LOOCV | GLM | 0.514 | 0.507 | 0.815 | 0.803 | 0.386 | 0.379 |
| | GAM | 0.519 | 0.512 | 0.818 | 0.812 | 0.388 | 0.381 |
| | BRT | 0.518 | 0.509 | 0.811 | 0.803 | 0.386 | 0.383 |
| | RF | 0.512 | 0.503 | 0.812 | 0.803 | 0.386 | 0.381 |
| CV | GLM | 0.517 | 0.505 | 0.815 | 0.807 | 0.379 | 0.371 |
| | GAM | 0.514 | 0.508 | 0.814 | 0.805 | 0.378 | 0.377 |
| | BRT | 0.513 | 0.504 | 0.815 | 0.811 | 0.381 | 0.371 |
| | RF | 0.521 | 0.516 | 0.818 | 0.813 | 0.383 | 0.376 |
| Random Splitting | GLM | 0.555 | 0.543 | 0.811 | 0.804 | 0.387 | 0.382 |
| | GAM | 0.541 | 0.532 | 0.816 | 0.811 | 0.385 | 0.378 |
| | BRT | 0.543 | 0.528 | 0.813 | 0.802 | 0.386 | 0.382 |
| | RF | 0.548 | 0.533 | 0.813 | 0.803 | 0.384 | 0.378 |

*3.5. Ensemble Model Development and Prediction*

Table 8 showed the predictive performances of all the single-algorithm models based on AIC, RMSE, and MAE using 100% of data. All the four single-algorithm full models exhibited proximal and favorable performance. The GAM-, BRT-, and RF-derived full models were selected and the GLM-derived full model was excluded because P.CPUE obtained through this model exhibited the highest AIC, RMSE, and MAE value of all

4 single-algorithm models with S.CPUE. Hence, final ensemble model was constructed using only GAM, BRT, and RF single-algorithm models. Model validation showed no significant differences in R, RMSE, and MAE values between training and test data for ensemble modeling all the validation techniques applied. Figure 4 reveals the findings as follows. From March to September, a high immature albacore S.CPUE was mainly observed between 30°S and 40°S; after September, a northward shift occurred; and from November to January, S.CPUE was mainly concentrated at the south coast of Madagascar. After January, an eastward shift occurred. The P.CPUE obtained from the ensemble model exhibited a high correlation with S.CPUE (0.867). A random splitting evaluation did not reveal any significant bias.

**Table 8.** Evaluation of single-algorithm models with full data set for ensemble model construction.

| Single-Algorithm Model | AIC | RMSE | MAE |
|---|---|---|---|
| GLM | 13,254.23 | 0.771 | 0.456 |
| GAM | 11,354.15 | 0.623 | 0.402 |
| BRT | 10,999.87 | 0.598 | 0.376 |
| RF | 10,785.35 | 0.595 | 0.354 |

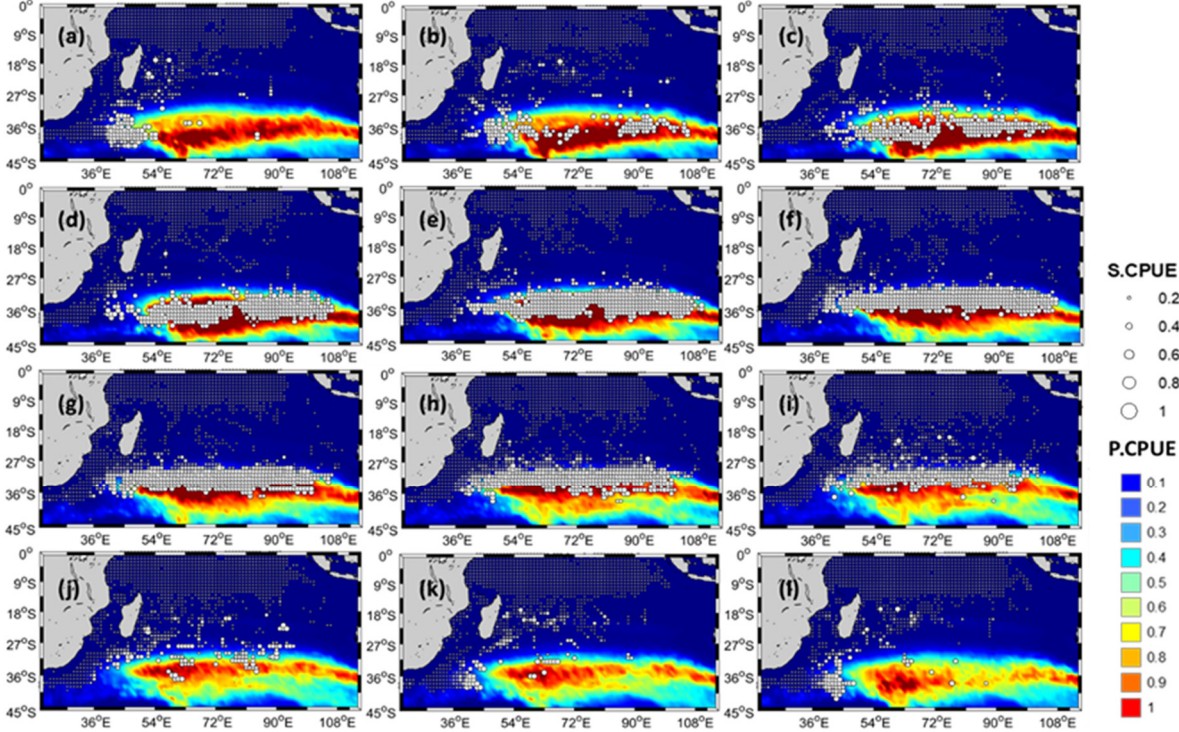

**Figure 4.** Monthly three-dimensional spatial distribution of immature albacore tuna in Indian Ocean from 1998 to 2016. (**a**) January, (**b**) February, (**c**) March, (**d**) April, (**e**) May, (**f**) June, (**g**) July, (**h**) August, (**i**) September, (**j**) October, (**k**) November, (**l**) December.

## 4. Discussion

The present study proposed a new method for inferring the potential habitats of immature albacore tuna, a commercially and ecologically crucial resource, in the Indian Ocean, on the basis of vertical oceanographic preferences. We used a combination of state-of-the-art species habitat algorithms and three-dimensional oceanographic data obtained from a high-resolution numerical model to develop high-quality habitat projections for immature albacore tuna. We investigated vertical immature albacore habitat characteristics, the potential link between environmental circumstances and potential albacore habitats by

applying the aforementioned method. Consequently, we examined the ensemble's habitat models quantitatively for operational albacore resource management applications.

### 4.1. Evaluation of Ensemble Model

Although the single-algorithm models exhibited consistent oceanographic parameter rankings, they differed in their intermodel statistical performance. These findings reflect an inherent intermodel uncertainty, indicating that the predictive power of statistical methods varies significantly [9]. Furthermore, compared with the top-performing single-algorithm predictions, combining immature albacore habitat forecasts by using the weighted means of three of the four top-performing models resulted in improved performance with respect to the accuracy of immature albacore habitat predictions (Figure 4). This finding corresponds to those of other studies, which reported that using the ensemble model method instead of a single-algorithm platform to forecast marine habitat affinities produces more favorable results [34,47]. Therefore, the ensemble model technique can be a useful tool for operational fisheries (e.g., mapping probable fishing areas) and resource management (e.g., identifying predictable foraging habitats that may include key zones for fish). Three possible reasons can be given in order to describe the good prediction failure by ensemble model in the panel a-j-k-l. Firstly, noise, bias and variance: the combination of decisions from multiple models can help to improve the overall performance. Hence, one of the key reasons to use ensemble models is overcoming noise, bias and variance. However, ensemble models in machine learning might have any noise, bias and variance from different models. This can be one possible reason for few prediction failures in the present study. Authors planned to use more advanced models for ensemble prediction in order to rectify areas of lack in the present study. The second reason was the existence of selected ideal environmental ranges. Furthermore, the area with a high S.CPUE of immature albacore tuna from April to September is where the fish feed. These two details can be used to map out where young albacore tuna are found in the Indian Ocean. Since October through January are not prime feeding months, and because it is summer, young albacore may vertically dive to reach the best SST. These reasons might be the causes of the decline in high CPUE values observed in the southern region between October and January.

### 4.2. Immature Albacore Habitat

In water temperatures between 11.5 °C and 18 °C, albacore tuna use physiological thermoregulation to maintain a stable body temperature of approximately 20 °C [48]. The body temperature of albacore tuna has been reported to decrease to <11.5 °C, which prompts individual tuna fish to change their behavior by migrating vertically into warmer waters to restore their body temperature. During their active growth phase, albacore tuna require a considerable amount of energy, which may explain their presence in areas where prey congregate at the water surface. Few studies have explored the high-energy intake of immature albacore tuna [49,50]. The high-energy requirements of such fish are expected to affect their distribution; specifically, individual fish tend to congregate in high-productivity locations where food is abundant. Because of the abundance of prey species in surface waters and the minimal need to forage in deeper waters, the vertical distribution of albacore tuna may be restricted to surface waters. However, with their thermoregulation capabilities [48] and swim bladder development [51], albacore tuna can dive into deep waters [52]. Our findings suggest that immature albacore tuna engage in vertical diving behavior only when necessary (e.g., for forging, hiding, thermoregulation, and osmoregulation). Moreover, our results indicate that the vertical distribution of immature albacore is affected by thermal preferences (as indicated by their diet); the vertical distribution of prey species, which is influenced by the ocean's thermal structure; or a combination of both factors.

Results indicated that, of the examined layers, the temperature at 5 m depth was the most significant, followed by that at 26 m depth. With a VIP score of 7, temperature at 53 m of depth did not demonstrate any major importance. At 147 m, oxygen ranked third in importance, ahead of 200 and 244 m. This occurred because 508 m depth was

more significant than 411 m and 628 m depth for both the SSS and SSC1, respectively. With a VIP ranking of 13, MLD displayed the lowest correlation with S.CPUE. In terms of the maximum S.CPUE, the best OXY range for both the 200 and 147 m depths was 240–260 mmol/L, whereas it was slightly lower for the 244 m depth (235–255). For the highest S.CPUE, the best TEM temperature ranges at 5, 26, and 53 m deep were 13–15 °C, 12–14 °C, and 14–16 °C, respectively. The ideal SSS range was 34.3–34.4 psu at both 508 and 628 m depth, but this range was slightly greater at 411 m depth (34.4–34.5 psu). For MLD, a range of between 240 and 260 m produced the highest S.CPUE. Changes in SST will control the muscle contraction. Slower muscle contraction will reduce the tail beat frequency and eventually will affect the swimming speed of fish. With the increase in SST, maximum swimming speed increases. However, an increase in SST greater than the preferred range will increment the energy costs for albacore, resulting in diminished execution and push, and it will influence the growth of albacore. Ambient SST affects the maximum swimming speed of fish. Being a carnivorous species [1], albacore has an indirect relationship with chlorophyll. SSC is the primary producer in the oceanic ecosystem, with secondary producers such as fish [4,6], shrimp, squid, and octopus feeding on it. Immature albacore then feed on these secondary producers. Thus, SSC is a critical factor to consider when describing the abundance of immature albacore tuna. SSS can be a crucial predictor. If the SSS is decreased from the preferred range of albacore then water density will also decrease, eventually affecting the swimming behavior (swimming is difficult in less dense water, thus swimming will cost higher energy loss) of albacore tuna. If the SSS is higher than the preferred range of immature albacore, then it will affect the osmoregulatory cost (use of extra energy) and begin affecting the growth of albacore tuna. MLD and SSH are related to each other and to SST. MLD is primarily determined by the action of turbulent mixing of the water mass due to wind stress and heat exchange at the air–sea interface. Cooling in SST can induce convection, which enlarges the MLD and decreases SSH.

Two possible reasons can affect the higher S.CPUE (used as the proxy of relative abundance) of immature albacore tuna. Presence of preferred optimal environmental ranges can be the first reason. Moreover, the location with high S.CPUE of immature albacore during April to September is the feeding zone of immature albacore tuna. These two facts can describe the spatial distribution of immature albacore tuna in the Indian Ocean. October to January is not the feeding time, and because of summer, an increase in SST might cause immature albacore to vertically dive in order to obtain the optimal SST. These might be the reasons behind the disappearance of high CPUE values in the southern sector from October–January.

### 4.3. Potential Implications for Albacore Fisheries

Understanding the species' regional dynamics and interactions at the population scale in all oceanic regions is a key aspect of efforts to improve albacore management. Because albacore movement is affected by seasonal (ideal temperature) and food supply considerations, population structures found between and within oceans are challenging to explain [53]. Furthermore, albacore tuna shift vertically and horizontally in water columns throughout their life cycle, a behavior which increases the difficulty of determining their distribution. Multiple species distribution models, stratified by space and time for each ocean basin and albacore tuna fishery, are required to clarify the albacore's complex distribution and spatial dynamics and the effect of changing environmental conditions.

### 5. Conclusions

In addition to the surface variable, the most influential vertical SST, OXY, SSC1 and SSS layers were found at various depths (i.e., 5, 26, and 53 m for SST; 200, 244, and 147 m for OXY; 508, 628, and 411 for SSCI; and 411, 508, and 773 m for SSS). Relative to the upper-surface base models, these layers had the lowest AICs. The GLM, GAM, BRT, and RF methods produced S.CPUE–P.CPUE correlation values of 0.798, 0.832, 0.841, and 0.856, respectively. The anticipated CPUE obtained through the GLM-derived full model exhibited the lowest

correlation with S.CPUE; thus, only the GAM-, BRT-, and RF-derived full models were selected. From March to September, a high level immature albacore S.CPUE was mainly observed between 30°S and 40°S; after September, a northward shift occurred; and from November to January, S.CPUE was mainly concentrated at the south coast of Madagascar. Our findings indicate that the vertical distribution of immature albacore is affected by thermal preferences (as indicated by their diet); the vertical distribution of prey species, which is affected by the ocean's thermal structure; or a combination of both factors.

**Author Contributions:** Conceptualization, S.M. and M.-A.L.; methodology, M.-A.L. and S.M.; software, S.M.; validation, M.-A.L.; formal analysis, S.M.; investigation, Y.-C.W. and J.-S.W.; resources, Y.-C.W. and J.-S.W.; writing—original draft preparation, S.M.; writing—review and editing, M.-A.L. and B.K.M. Mondal, visualization, Y.-C.W. All authors have read and agreed to the published version of the manuscript.

**Funding:** The Council of Agriculture (106AS-10.1.5-FA-F1 (4); 107AS-9.1.5-FA-F1 (4)) and Ministry of Science and Technology of Taiwan (MOST105-2611-M-019-011) financed this research.

**Data Availability Statement:** Not applicable.

**Acknowledgments:** We thank the three anonymous reviewers and the editor for their valuable comments and suggestions as well as the team members of the Fisheries Agency and Overseas Fisheries Development Council of Taiwan for their assistance in data preparation. We are grateful to the Council of Agriculture, Taiwan, for the grant.

**Conflicts of Interest:** The authors declare no conflict of interest.

## Abbreviations

| | |
|---|---|
| SDM | Species distribution model |
| MODIS | Moderate Resolution Imaging Spectroradiometer |
| COP | Copernicus |
| AVHRR | Advanced Very High Resolution Radiometer |
| SST | Temperature |
| OXY | Dissolved oxygen |
| SSC (0–2) | Chlorophyll (0–2 months lag) |
| SSS | Salinity |
| U | U-velocity |
| V | V-velocity |
| EKE | Eddie kinetic energy |
| NPP | Net primary productivity |
| MLD | Mixed layer depth |
| SSH | Sea surface height above geoid |
| N.CPUE | Nominal catch per unit effort |
| S.CPUE | Standardized catch per unit effort |
| GLM | Generalized linear modeling |
| AIC | Akaike information criterion |
| $R^2$ | Correlation |
| GAM | Generalized additive model |
| BRT | Boosted regression trees |
| RF | Random Forest |
| RS | Random splitting |
| LOOCV | Leave-one-out cross-validation |
| R | Pearson correlation coefficient |
| RMSE | Root mean square error |
| MAE | Mean absolute error |
| P.CPUE | Predicted catch per unit effort |

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
