# Peer review of "Ensemble Three-Dimensional Habitat Modeling of Indian Ocean Immature Albacore Tuna (Thunnus alalunga) Using Remote Sensing Data"

_remotesensing, doi:10.3390/rs14205278_

Round 1

Reviewer 1 Report

In this manuscript, the authors used the ensemble model technique and 3D oceanographic data to evaluate the  habitats of immature albacore tuna in the Indian Ocean

Overall, the manuscript is well written and structured. It can be accepted if the following issues can be further clarified.

(1) The 3D oceanographic data was obtained form the existing database. based on my reading, this comes from the Oversees Fisheries Development council of Taiwan. This manuscript seems to be a data analysis of the existing data. What is the real new scientific contribution to the field.

(2) How the technique is related to the scope of this journal? I don't see the relevant information with sensor.

(3) In the model component, the authors used four models , GLM, GAM, BRT, and RF for the analysis. The pack-180 age’s species distribution models also use a set of statistical techniques, which include 181 conventional regression-based models and current machine-learning platforms. These models are conventional technique. What is new contribution form this manuscript is unclear.

In summary, the authors need to clearly state the new scientific contribution of this manuscript. Otherwise, it is only an exercise for data analysis of the existing data, not a scientific research.

Author Response

Kindly refer to the attached word file named "Reviewer_1".

Reviewer 2 Report

General comments

This paper explores the relationship between the spatio-temporal distribution of the CPUE of immature albacore tuna and environmental parameters using interesting methods for the data analysis. From an analytical point of view, this work is interesting as it goes beyond the use of single algorithms and benefits from both multivariate statistical models and machine learning techniques through an ensemble method.

The descriptive part of the materials and methods is not very detailed and some inaccuracies should be improved. Although I failed to understand some parts of the methods (at least until I read the results), I believe that the general conduction of the analysis is correct.

From a biological point of view, I believe that the work is very weak. In fact, this part does not adequately value the effort made to implement the analysis, making this methodological effort an end in itself. The only biological part supported by the data present in this work concerns the spatial and temporal distribution of the CPUE, which could also be done without the use of models. In my opinion, a description of the relationship between variable response and covariates that could trigger a discussion on the ecology of this juvenile stage is totally missing. For example, which of the chosen variables are most important? For which values of these variables can higher CPUEs be associated? See the case study of the anchovy described in Quinci et al. (Water, 2022, 14 (9), 1400) for a good approach regarding this argument.

Finally, I think it is speculative to consider this work as able to describe the vertical distribution of albacore tuna's juveniles. In fact, in this work, there is no data relating to the depth of capture of individuals and this could betray the reader's expectations. Indeed, what has been explored in this work is the relationship between the values of the environmental parameters along the water column and the CPUE. I therefore consider the use of the first phase of the abstract "This study evaluated the vertical distribution of immature albacore tuna (Thunnus alalunga) in the Indian Ocean as a function of various environmental parameters" to be wrong. Similarly, some parts of the discussion concerning the ecological aspect are speculative and not supported by the results described. I believe the interpretation of model outcomes as well as the discussion on biological and ecological aspect of this important species could be improved.

For all these reasons, about this work I recommend major revisions before being re-evaluated for acceptance in this journal. Further suggestions are reported in the specific comments.

Specific Comments

Abstract

Lines 23-26. It is not clear why that layers are the “most influential”. Please, consider to review this sentence adding this information, otherwise I can suggest to delete it because is referred to a specific part of the methodology. Moreover, you should specify if you are talking about daily, monthly or yearly data.

Line 26. Why “upper-surface”?

30-32. It is not clear why you are doing a selection of your models. What is the purpose of this selection? Moreover, why you exclude the lowest in terms of correlation between observations and predictions? Can you find a more objective criteria, such as a threshold, in order to perform your selection?

Introduction

Lines 41-43: The references 4 and 5 are dated. Please provide a more recent percentage value of the global catches

Lines 67-68: references 13 and 14 are dated. You could provide a more recent references on this topic, e.g. I could suggest:

Russo et al. (2021). Unveiling the Relationship Between Sea Surface Hydrographic Patterns and Tuna Larval Distribution in the Central Mediterranean Sea. Frontiers in Marine Science, 2021, 8, 708775

Russo et al. (2022). Environmental Conditions along Tuna Larval Dispersion: Insights on the Spawning Habitat and Impact on Their Development Stages. Water (Switzerland), 2022, 14(10), 1568.

Lines 75-76: You could cite a more recent references on this topic, e.g. I could suggest:

Quinci et al. (2022). Predicting Potential Spawning Habitat by Ensemble Species Distribution Models: The Case Study of European Anchovy (Engraulis encrasicolus) in the Strait of Sicily. Water (Switzerland), 2022, 14(9), 1400

Materials and Methods

In general, this section is not very detailed. In general, a description of the distributions, functions and parameters used in the models is missing. For example, in GLM and GAM there is no information about the assumed distribution (Gaussian? Gamma?) And the link function used. In addition, the type of spline regression used is missing. The RF does not report the number of trees used and the number of variables used in each split point (i.e. mtry) and so on. Please consider to add a specific part of the methods that can report these information.

Lines 108-113. Please, double check if tonnage and length data about small and large vessels are correct.

Lines 125-130: I believe you should be more precise about the methods. What spatial interpolation method did you use (IDW, kriking etc ...)? Did you use averages to convert dalily data to montly data? Please specify them.

Lines 140-150. Before reading the results, I failed to understand in the methods how you standardized CPUE data. In particular:

·        “The mgcv package was used to build a stepwise GLM model with one to five factors”: What kind of factors are you considering in this step? Are you talking about spatial (lat and lon) and temporal (year and moth) factors? What is the fifth covariates?

·        “μ is the interaction (Year*Lat, Year*Lon, and Lat*Lon)”. How are these 3 factors distributed in the 5 models? All together?

Even if in the table 1 (performances) are reported the five models, in this part is not really easy to understand what did you do. Please, revise this part and be more specific about the five implemented models.

Line 141: is N,CPUE or N.CPUE? Please revise it.

Lines 156-158: The references are dated. You could cite a more recent references on this topic, e.g. I could suggest:

Patti et al. (2022). Interannual summer biodiversity changes in ichthyoplankton assemblages of the Strait of Sicily (Central Mediterranean) over the period 2001–2016. Frontiers in Marine Science, 9, 960929

Torri et al. (2021) Signals from the deep-sea: Genetic structure, morphometric analysis, and ecological implications of Cyclothone braueri (Pisces, Gonostomatidae) early life stages in the Central Mediterranean Sea. Marine Environmental Research, 2021, 169, 105379.

Lines 162-164. What kind of correlation are you talking about? Please consider to provide more information on this aspect of the methodology. I guess you are talking about the R2. Please specify it.

Lines 168: Did you use the R Person coefficient for the evaluation of the collinearity between covariates? Why don’t you used VIF? Please explain better and justify your choice.

Lines 172-173. Why you exactly selected these layers? Does it depend on the type of product downloaded from copernicus? Are you used a model-derived product (as I guess) or a satellite observations? I believe you should better specify the metadata of the CMEMS product you used for your analysis

Line 186. Is [LOO] CV or [LOOCV]? Please revise it.

Lines 189-194 and lines 201-205: Why you calculated the three coefficients (R, RMSE and AE) in both dataset (training and testing)? Usually, the training dataset is used to train the model, while the testing is used to compare the model performances (e.g. applying the trained model to the testing dataset and comparing predictions vs testing observations). I did not understand what is the advantage of dividing the dataset into two random parts (70:30) and then applying the same analysis to both. In fact, the analysis thus made does not justify the use of the terms "Training" and "Testing". Please explain better.

Line 193: is MAE or AE? Please revise it.

Results

Lines 257-259: What you used doesn't seem to be a test but a simple correlation coefficient. Differently, you could apply a test in order the assess the significance of the correlation coefficient you calculated (e.g. cor.test function in R). Otherwise, you could use the VIF. Please revise it, as well as the table 4.

Lines 272-278. Again, I failed to understand why you splitted your dataset in training and dataset and then you applied the validation technique to both. For instance, if you apply the random splitting to your testing dataset (that is already the 30% of your original dataset), you are splitting again this dataset in other two subset. These considerations are valid also with the other three techniques. In my opinion, this does not lead to any apparent advantage but it would seem a mistake, because the algorithm of all these techniques already performs a division of the original dataset into several subsets in order to perform an assessment of the performances. Please explain better why you used this approach, otherwise use the training and the testing dataset appropriately.

Lines 283-284. Where is it evident that the GLM obtains the lowest correlation between S.CPUE and P.CPUE? Please, explain better how you support this statement.

Lines 284-290: can you provide an explanation of why your ensemble model fail to make a good prediction in the panel a-j-k-l of the figure 4?

Lines 285-288. Why you talk about a “shift”. From the figure 4 it seems evident a quite stable CPUE in the northern part of the study area in all months of the year. How can you exclude that the disappearance of high CPUE values in the southern part is not the result of a movement of these individuals outside the study area? Please, consider to discuss this aspect in the discussion section.

Discussion

Lines 313-315. For a more recent paper on this topic, I can suggest the already mentioned paper Quinci et al., 2022.

Lines 338-340- This is actually a speculation as you didn’t evaluate. Please support this sentence with your results or delete this sentence.

Lines 340-341. Please explain how your result support this sentence. Actually, your analysis did not evaluated parameters concerning depths greater than 1045 meters. Moreover, you didn't show any information about the depth at which the immature fish were caught.

Lines 341-346. This is a pure speculation. Indeed, your finding doesn’t support this statement. Please, consider to show how your data support this sentence, otherwise delete it.

Lines 368-371. This is a pure speculation. Indeed, your finding doesn’t support this statement. Please, consider to show how your data support this sentence, otherwise delete it.

Figure 4. Are the different panels (a-l) referred to the different sampling months? Is a=January, b=February and so on? You could better specify it in the caption and/or adding a title for each panel. Moreover. Are data collected in the entire period (1998-2016) plotted together? Please specify it. Moreover, the difference in size of the circles referred to S.CPUE in not really appreciable. Please modify it.

Table 1 (sources): Please specify if you are using L3 or L4 data. Moreover, you should specify if you are using observations or model-derived data. Moreover, why you included Clorophyll-a concentration with 1 and 2 months of time lag? There is a biological explanation? I believe its very important specify why you selected these parameters in Material and Methods.

Table 2: I believe there is an error in the numeration of this table.

Table 5: Is the validation technique [repeated 10-f] missing in the table? Moreover, you didn’t specify what the CV is. You should better specify them in the caption.

Figure 1: About environmental data collection, Copernicus, Modis e Aviso sources are indicated. However, table 1 report only product downloaded by CMEMS. Please double check and, in case, explain better the sources used in this paper.

Keywords: In the keywords, 'vertical' or “ensemble” alone doesn't make much sense. I would choose keywords that are more impactful (e.g. ensemble distribution modelling).

Author Response

Kindly refer to the revised manuscript with track changes and for the reply to the comments please refer to the file titled "Reviewer_2"

Reviewer 3 Report

This study evaluated the vertical distribution of immature albacore tuna (Thunnus alalunga) in the Indian Ocean as a function of various environmental parameters for the period from 1998 to 2016.

Following the finalization of the oceanographic variables based on lowest Akaike information criteria, four single-algorithm models were built: the generalized linear model (GLM), generalized additive model (GAM), boosted regression tree (BRT), and random forest (RF) model. The correlation values between the standardized and predicted catch per unit effort (CPUE) for the models were 0.798, 0.832, 0.841, and 0.856, respectively. The GAM-, BRT-, and RF-derived full models were selected, whereas the GLM-derived full model was excluded because its correlation value was the lowest among the four models.

The work is well written, it compares different distribution models, the applied techniques are valid and the results well described. I find the references part missing, many are in the text inserted after the explanation (eg [45], others are missing for some statistical methods or packages used. The tables must be numbered correctly in the text and in the description. Many acronyms are used before definition.

·         Many of the syntax were not well defined, such as SSCI, SSS, SSCI, etc. These acronyms should be defined when they first appear, instead in the middle of text.

Maybe the authors can add a list of all acronyms used in the text.

·         Li 141 N,CPUE is N.CPUE?

·         Li 144 Insert reference for mgcv package R

·         Li 145 Akaike information criteria: need explanation of method, the reference [45] was in line 175.

·         The references list must improve and used in the first  

·         Li 224 Table 1 appear two times, renumber.

·         Li 275 R2 is R2?

·         Figure 4. The panels are indicated as (a),(b)… but in the text (li 365-368) the authors referred directly to the months. Use in the panels (JAN), (FEB)…ecc  

Author Response

Kindly refer to the attached word file named "Reviewer_3".

Reviewer 4 Report

This manuscript clearly shows the research objective, gap, novelty, and its clear research design. In general, this manuscript is accepted for publication. However, I have a few comments that the authors need to address:

1.      How to separate the dataset of immature and mature albacore tuna during the dataset collection. I think better to put the detail in the data and method section.

2.      Why the authors used Standardization of CPUE instead of CPUE?

3.      The authors mentioned 70:30 (training: testing), Is it a ratio? Please clarify the sample number as well. 

Author Response

Kindly refer to the attached word file named "Reviewer_4".

Round 2

Reviewer 1 Report

Thanks the authors devoted the great efforts to revise the manuscript. The quality of the manuscript has been significantly improved. In my opinion, the manuscript can be accepted with the present form.